# Emerging Roles of the Selective Autophagy in Plant Immunity and Stress Tolerance

**DOI:** 10.3390/ijms21176321

**Published:** 2020-08-31

**Authors:** Jie Ran, Sayed M. Hashimi, Jian-Zhong Liu

**Affiliations:** 1College of Chemistry and Life Sciences, Zhejiang Normal University, Jinhua 321004, China; ranjie15736516243@163.com (J.R.); s.masoud.hashimi@gmail.com (S.M.H.); 2Zhejiang Provincial Key Laboratory of Biotechnology on Specialty Economic Plants, Zhejiang Normal University, Jinhua 321004, China

**Keywords:** selective autophagy, autophagosome, xenophagy, cargo receptor, plant immunity, abiotic stress, ubiquitin–proteasome system, 26S proteasome

## Abstract

Autophagy is a conserved recycling system required for cellular homeostasis. Identifications of diverse selective receptors/adaptors that recruit appropriate autophagic cargoes have revealed critical roles of selective autophagy in different biological processes in plants. In this review, we summarize the emerging roles of selective autophagy in both biotic and abiotic stress tolerance and highlight the new features of selective receptors/adaptors and their interactions with both the cargoes and Autophagy-related gene 8s (ATG8s). In addition, we review how the two major degradation systems, namely the ubiquitin–proteasome system (UPS) and selective autophagy, are coordinated to cope with stress in plants. We especially emphasize how plants develop the selective autophagy as a weapon to fight against pathogens and how adapted pathogens have evolved the strategies to counter and/or subvert the immunity mediated by selective autophagy.

## 1. Introduction

Macroautophagy, referred to as autophagy, is an evolutionary conserved pathway that engulfs the damaged or no longer needed cytoplasmic components to double membrane vesicles called autophagosomes [1]. Under normal growth conditions, autophagy helps cells maintain metabolite homeostasis, whereas under stress conditions, it is activated to degrade damaged organelles or protein aggregates for nutrient recycling [2,3,4,5]. Autophagy has been shown playing critical roles in a wide range of physiological processes ranging from growth and development, stress adaptations, cell survival and death, as well as disease resistance [2,3,6,7,8,9,10,11]. Autophagy was initially considered to be a nonspecific catabolic process, which was termed as bulk autophagy. However, it is now clear that particular cargoes can be degraded specifically by selective autophagy in response to diverse stress conditions [2,9,10,12,13,14].

Ubiquitin-like Autophagy-related gene 8 (ATG8) plays critical roles in selective autophagy. ATG8 proteins are anchored in both the inner and outer membranes of autophagosomes by a conjugation pathway, which attaches the lipid phosphatidylethanolamine (PE) to its carboxyl terminus [15]. The membrane-anchored ATG8 not only provides a docking platform for the ATG8-interacting proteins that are essential for phagophore initiation, elongation, and maturation, but also for the recruitment of cargoes selectively mediated by cargo receptors [16,17]. The cargoes targeted by selective autophagy are recognized by cargo receptors that interact with membrane-anchored ATG8 through their ATG8-interacting motifs (AIM), which contain the consensus core W/Y/F-XX-L/I/V sequence [16,18]. The AIM forms hydrophobic bonds with two conserved hydrophobic pockets of the AIM-docking sites (ADS) on ATG8s [16]. The three-way interactions lead to the recruitment of the cargoes to the autophagosomes, thereby facilitating their delivery to lysosomes/vacuoles for degradation [17,19]. Recently, Marshall et al. [20,21] identified a new class of selective receptors that interact with ATG8s through an ubiquitin interacting motif (UIM). UIM bind to ATG8s on an UIM-docking site (UDS) with high affinity [10,20]. The UDS is not only present in the Arabidopsis ATG8 homologs, but also in ATG8 orthologs from yeast and animal, suggesting that the selective autophagy mediated by UIM–UDS is conserved across kingdoms [20].

The list of ATG8-binding proteins has increased substantially in the past few years [20,22] and the importance of selective autophagy has become apparent (Figures 1 and 2; Tables 1 and 2). It has been demonstrated that the selective autophagy plays central roles in removing protein aggregates [23,24,25] as well as the damaged organelles such as plastids (chlrophagy) [26,27,28,29], peroxisomes (pexophagy) [30,31,32,33], mitochondria (mitophagy) [34], endoplasmic reticulum (ER) (ER-phagy) [35], and ribosomes (ribophagy) [36,37,38] under various stress conditions. Recently, Marshall et al. [21] showed that inactive 26S proteasomes are removed via a mechanism called proteaphagy, establishing a functional link between the two major degradation pathways. Furthermore, selective autophagy also plays a critical role in the clearance of invading pathogens (xenophagy) including bacteria, viruses, and fungi [11,13,39,40,41]. Since numerous excellent reviews have covered the pexophagy [42,43], chloropahgy [28,44,45], ER-phagy [3], proteaphagy [21], and xenophagy [11,13,41,46,47], we just focus our attention on the newly emerged features and the novel roles of the selective autophagy in response to both biotic and abiotic stresses in plants.

## 2. Universal Receptors Directly Mediate the Selective Autophagy of Ubiquitinated Proteins under Various Stress Conditions

Misfolded and damaged proteins under various stress conditions are highly toxic to the cells. Plants develop highly sophisticated and efficient strategies to repair, refold, or degrade these damaged proteins. Protein aggregation occurs when chaperone-mediated refolding and/or proteasome degradation are overwhelmed by an excessive amount of misfolded proteins under various stress conditions [48]. The clearance of stress-induced ubiquitinated proteins aggregates and/or damaged organelles via selective autophagy is a common strategy for plants to tackle various stresses.

### 2.1. Neighbor of BRCA1 (NBR1) Mediates Selective Autophagy of Polyubiquitinated Proteins or Protein Aggregates under Various Stress Conditions

Plant NBR1, a homolog of mammalian autophagy cargo adaptors p62 and NBR1, is the best characterized universal cargo receptor that mediates the degradation of polyubiquitinated protein aggregates (aggrephagy) as well as pathogen proteins or even entire pathogens (xenophagy) [25,49,50]. NBR1 homologs contain both a UBA (ubiquitin-associated) domain and AIM, allowing them to recruit ubiquitinated cargoes to the ATG8-labeled autophagosomes [50,51,52,53]. In Arabidopsis, AIM is indispensable for the function of *At*NBR1, indicating that it functions as a receptor for selective autophagy [23,25]. *At*NBR1 mutants are highly sensitive to various biotic and abiotic stress conditions [23,25,40,54,55]. The reduced tolerance of *nbr1* mutant to various abiotic stresses is highly correlated with the enhanced accumulation of ubiquitinated proteins and/or insoluble detergent-resistant and/or protein aggregates [23,24,25,54]. The NBR1-mediated selective autophagy does not appear to target specific proteins. Instead, it more likely targets ubiquitinated cellular proteins under various stress conditions [25].

### 2.2. Selective Autophagy Collaborates with a Ubiquitin–Proteasome System (UPS) to Deal with Various Stress

UPS and autophagy are the two major degradation pathways. Recent studies showed that these two major degradation systems collaboratively fight against stress by clearing misfolded proteins or protein aggregates induced under stress conditions [24]. Carboxyl terminus of the Hsc70-interacting protein (CHIP), a chaperone-associated E3 ubiquitin ligase, is responsible for the degradation of a number of light-harvesting complex proteins by 26S proteasomes, while NBR1 preferentially and selectively mediates the autophagic degradation of the highly aggregate-prone proteins such as rubisco activase and catalases, indicating that CHIP and NBR1 mediate two distinct but complementary anti-proteotoxic pathways to deal with stress [24]. Rubisco activase and other chloroplastic proteins might have been ubiquitinated by CHIP prior to being targeted for autophagic degradation and NBR1 mediated their recruitment to the autophagosome through its UBA domain.

BRI1-EMS suppressor 1 (BES1) is a positive regulator in the Brassinosteroid (BR) pathway that promotes plant growth [56]. The transcription activity of BES1 is negatively regulated by a Glycogen synthase kinase 3 (GSK3)-like kinase Brassinosteroid-insensitive 2 (BIN2) [57], whose activity is induced under stress conditions [58,59,60]. In the absence of BRs, BIN2 phosphorylates and inhibits BES1 function [61]. However, in the presence of BR, the kinase activity of BIN2 is inhibited, leading to the accumulation of dephosphorylated BES1 in the nucleus to regulate the expression of downstream target genes [62,63]. Dominant suppressor of KAR 2 (DSK2) is an ubiquitin-binding receptor protein that participates in the delivery of ubiquitinated cargo proteins to the proteasome for degradation [64,65]. Nolan et al. [66] recently showed that in response to stress, DSK2 interacts with both the ubiquitinated BES1 mediated by Seven in absentia of Arabidopsis 2 (SINAT2), a Really interesting new gene (RING) E3 ligase, and ATG8e, mediating the autophagic degradation of BES1. DSK2A contains two putative AIMs in its amino acid sequence, each of which is flanked by multiple consensus phosphorylation sites of BIN2 [67]. The phosphorylation of DSK2 by BIN2 increases the interaction of BES1 with ATG8e and accelerates its autophagic degradation under drought conditions [66]. The degradation of BES1 leads to an altered global transcriptome, which inhibits Arabidopsis growth/development and activates stress responses.

These results reveal a novel mechanism by which plants balance growth and stress responses by targeting a central growth regulator to the selective autophagy pathway via a phosphorylation-regulated receptor protein [66]. As DSK2 contains a UBA domain, it recruits BES1 to autophagosomes likely through its UBA domain by interacting with the ubiquitin chains on BES1 catalyzed by SINAT2. If this is true, DSK2 could serve as a selective receptor to mediate the autophagic degradation of additional uqbiquitinated proteins under various stress conditions. It remains to be determined whether a subset of those ubiquitinated cargo proteins mediated by DSK2 for proteasome degradation is also degraded by the selective autophagy pathway.

In response to dehydration, a stress-inducible and ER-localized E3 ubiquitin ligase Rma1H1 targets the aquaporin PIP2;1 for proteasomal degradation [68]. Recent reports indicated that Aquaporin PIP2 proteins are also subjected to degradation by selective autophagy in response to water stress [69,70,71]. Turnip mosaic virus (TuMV) mediates Remorin1.2 (REM1.2) degradation via both the 26S proteasome and autophagy pathways [72]. The VPg (Viral protein genome-linked) of potyvirus, a potent RNA-silencing suppressor, antagonizes host defense through targeting Suppressor of Gene Silencing 3 (SGS3), a key silencing player functioning in double-stranded RNA synthesis, for degradation by both UPS and autophagy pathways [73]. These results indicate that collaborative participation of the UPS and the selective autophagy pathways in stress tolerance is a common phenomenon.

UPS degrades the misfolded/damaged proteins under normal or mild stress conditions. However, under severe stress conditions, the ubiquitinated proteins form large aggregates that exceed the capacity of the UPS, and the UPS itself is overwhelmed or even damaged. Under such circumstances, the selective autophagy kicks in to remove both damaged proteasome [21] and protein aggregates [24,25,66]. Ubiquitination serves either as a binding site for the UBA-containing selective receptors such as NBR1 and DSK2 [23,24,66,74] or as a signal for vacuolar degradation via the endotytic pathway [23,74].

### 2.3. UPS and Selective Autophagy Are Responsible for the Degradation of Ligand-Activated and Non-Activated FLS2, Respectively

*Arabidopsis thaliana* recognizes bacterial flagellin through a 22 amino acid conserved epitope at its N-terminal region, flg22. Upon binding to the receptor-like kinase (RLK) Flagellin-sensing 2 (FLS2), flg22 induces the association of FLS2 with its co-receptor, BRI1-associated kinase 1 (BAK1), which activates the receptor complex to trigger downstream defense responses [75,76,77,78]. The degradation of receptor-like kinases plays a critical role in activating and/or de-activating the defense signals. To prevent the excessive and prolonged activation of defense responses, which is detrimental to plants, the activated RLK must be de-activated. Lu et al. [79] showed that upon flg22 perception, BAK1 interacts with and directly phosphorylates PUB12 and PUB13, which are two typical Plant Ubox (PUB) E3 ubiquitin ligases. The phosphorylations of PUB12 and PUB13 by BAK1 result in their association with *At*FLS2, and thus the subsequent polyubiquitination and degradation of *At*FLS2 by the 26S proteasome [79]. This study uncovers the molecular mechanism by which plants attenuate innate immune responses following the pattern recognition receptor (PRR) activation.

Under non-elicited conditions, non-activated plasma membrane (PM)-localized *At*FLS2 constitutively recycles between the PM and endosomes via a clathrin-dependent endocytic trafficking route [80]. Intriguingly, it has been reported recently that orosomucoid (ORM) proteins regulate the stability of non-activated FLS2 [81]. ORMs have been reported regulating sphingolipid biosynthesis through suppressing the activity of serine palmitoyltransferase (SPT), which is a key enzyme in the sphingolipid synthesis pathway [82,83]. In addition to participating in sphingolipid synthesis, ORM proteins interact with both FLS2 and the ATG8s via an AIM and act as a selective autophagy receptor to mediate the degradation of FLS2 [81]. These results indicate that ORM proteins serve as selective autophagy receptors for non-activated FLS2 to modulate plant immunity. A logic question is: are other non-activated PRRs similarly subjected to selective autophagy? If yes, why do plants not evolve a strategy to use a common receptor for the selective autophagy of PRRs for energy-saving purpose? In addition, it remains to be determined how the biosynthesis, recycling, and selective autophagic degradation of the non-activated FLS2 are regulated and coordinated, and what are the signals for these different processes?

### 2.4. Both a Linker Adaptor and a Selective Receptor Are Required for the Autophagic Degradation of Ubiquitinated Cargo Proteins under Stress Conditions

Zhou et al. [23] identified 3 related dicot-specific ATG8-interacting proteins (ATI3A, ATI3B, and ATI3C). The loss function of ATI3s compromises both plant heat tolerance and resistance to the necrotrophic fungal pathogen *Botrytis cinerea*. The functions of ATI3s in heat tolerance and disease resistance totally rely on their interaction with ATG8a mediated by the AIM [23]. Interestingly, two conserved ER resident UBAC2 proteins (UBA protein 2a/b), implicated in ER-associated degradation (ERAD), were found interacting with ATI3s [23]. The functional connection of ATI3 and UBAC2 with the autophagy pathway is further established by the fact that both proteins are delivered to vacuole for degradation in an autophagy-dependent manner under ER stress [23]. The authors propose that ATI3 and UBAC2 participate in plant stress responses by mediating the selective autophagy of specific as-yet identified ubiquitinated ER components (Figure 1). It is possible that ATI3 serves as a receptor by interacting with ATG8, and the UBAC2 serves as an adaptor by binding ubiquitinated ER proteins for autophagy degradation [23]. If this is the case, ATI3s–UBAC2s could mediate the autophagic degradation of a suite of ubiquitinated ER proteins under stress conditions. However, the vacuolar delivery of ATI3s and UBAC2s under ER stress has not been examined in the absence of either ATI3s or UBAC2s. Nonetheless, this study shows that both a receptor (ATI3) and an adaptor (UBAC2) are needed for the clearance of ubiquitinated ER proteins via the selective autophagy pathway.

### 2.5. Selective Autophagy in Degradation of ER and Plastid Proteins under Stress

Honig et al. [84] identified two closely related homologous plant-specific proteins, termed ATG8-interacting proteins 1 and 2 (ATI1 and ATI2) as autophagy cargo receptors. Both proteins contain two putative AIMs and interact with the *At*ATG8f or *At*ATG8h [84]. Under normal growth conditions, both ATI1 and ATI2 localize to the ER. However, upon carbon starvation, they associate with the mobile bodies derived from ER (termed ATI1 bodies) that move along the ER membrane network and transported into the vacuole, suggesting that they mediate the transport of specific ER components into the vacuole.

Besides ER localization, ATI1 is also located on bodies associating with plastids, which are detected mainly in senescing cells that exhibit plastid degradation or under carbon starvation condition [85]. ATI1 is involved in the selective autophagy of plastid proteins through interacting with both plastid proteins and ATG8f [85]. Nine plastid ATI1-interacting proteins are identified and the ATI1-mediated autophagic degradation is confirmed for Peroxiredoxin A (PrxA). ATI1 is involved in Arabidopsis salt stress tolerance possibly through the clearance of damaged plastid proteins. However, it is still unclear whether ATI1 fulfills its function through mediating chlorophagy (degradation of the entire chloroplast) or just the degradation of plastid proteins (piecemeal) [86].

### 2.6. Distinct Selective Receptors in Different Plant Species Target the Same Family of Proteins for Autophagic Degradation to Tackle Drought Stress

The Arabidopsis Multistress Regulator tryptophan-rich sensory protein/translocator (*At*TSPO) is a heme binding, early secretory pathway-localized membrane protein, whose expression is induced under heat and drought conditions [87]. Heme binding to *At*TSPO promotes its degradation via the autophagy pathway [69]. Using a split ubiquitin screening system, a plasma membrane-localized aquaporin PIP2;7 (Plasma membrane intrinsic protein 2;7) was identified as an *At*TSPO-interacting protein [70]. The aquaporins largely modulate the water flow across cell membranes [88,89]. Hachez et al. [70] showed that *At*TSPO reduces PM-localized PIP2;7 level in an autophagy-dependent manner, suggesting that *At*TSPO might function as a selective receptor to target PIP2;7 for autophagic degradation. The *At*TSPO-mediated degradation of PIP2;7 reduces the PM-localized PIP2;7 level and therefore limits PIP2;7-dependent water loss at the PM under osmotic stress conditions [70]. However, the direct evidence that *At*TSPO interacts with ATG8 and the importance of AIM of *At*TSPO in interacting with PIP2;7 and its functional relevance has not been shown in this study. In addition, the domains that are important for the interaction between *At*TSPO and PIP2;7 have not been identified. Intriguingly, *At*TSPO localizes in the early secretory pathway at ER, Golgi, and TGNs [69], but it mediates the autophagic degradation of a PM-localized aquaporin PIP2;7 [70]. It is not understood how this is achieved. One possibility is that instead of interacting with PIP2;7 at PM, *At*TSPO intercepts PIP2;7 at the ER, Golgi membranes, and TGNs, and it is targeted for autophagic degradation directly from these subcellular locations. This postulation is supported by the fact that *At*TSPO co-localizes and interacts with PIP2;7 in the ER and Golgi stacks [70].

Interestingly, Li et al. [71] recently showed that *Mt*CAS31 (cold acclimation-specific 31), a dehydrin, functions as a selective receptor for the autophagic degradation of aquaporin *Mt*PIP2;7 to modulate drought tolerance in *Medicago truncatula*. Dehydrins are classified as group 2 LEA (late embryogenesis abundant) proteins and exhibit both hydrophilic and hydrophobic characteristics, and they easily bind to biomolecules, such as nucleic acids, proteins, and membrane components [90,91]. *Mt*CAS31 interacts with both *Mt*ATG8a and *Mt*PIP2;7, respectively [71]. Under drought stress, *Mt*CAS31 facilitates the autophagic degradation of MtPIP2;7 and reduces root hydraulic conductivity, thus reducing water loss and improving drought tolerance [71].

It is intriguing that different proteins serve as receptors for the selective autophagic degradation of PIP2;7 homologs in different plant species. We wonder whether *Mt*TPSO and *At*CAS31 homologs can function reciprocally. Nonetheless, these results illustrate that on one hand, multiple receptors could mediate the autophagic degradation of the same cargo protein. On the other hand, the same receptor can mediate different cargo proteins for autophagic degradation.

### 2.7. Receptor Mediates Its Own Degradation via Selective Autophagy

S-nitrosoglutathione reductase 1 (GSNOR1) is a highly conserved master regulator of nitric oxide (NO) signaling through maintaining the intracellular level of *S*-nitrosoglutathione (GSNO), which is a major bioactive NO species and regulator of protein S-nitrosylation. Zhan et al. [92] showed that S-nitrosylation induces the selective autophagy of Arabidopsis GSNOR1 during hypoxia responses. Under hypoxia condition, GSNOR1 is S-nitrosylated at Cys^10^, and the S-nitrosylation of GSNOR1 at this site induces conformational changes, enabling its AIM to be exposed and accessible for ATG8 binding. This finding unravels a unique mechanism by which S-nitrosylation triggers the selective autophagy of GSNOR1.

The tomato AGC protein kinase AvrPto-dependent Pto-interacting protein 3 (Adi3) is known to function as a suppressor of programmed cell death (PCD) and the silencing of Adi3 leads to spontaneous cell death [93]. The ATG8h was identified as an Adi3-interacting protein through a yeast two-hybrid screening [94]. The silencing of genes involved in autophagy is known to lead to runaway PCD [6]. Co-silencing Adi3 with autophagy genes leads to the aggravated runaway cell death, suggesting that Adi3 may be involved in the autophagic regulation of PCD [94]. However, the autophagic degradation of Adi3 has not been shown in this study. It is unclear whether the ATG8h-Adi3 interaction mediates the degradation of Adi3.

The feature in these two cases is that these proteins serve as receptors and mediate their own degradation by the selective autophagy pathway. Usually, the selective autophagy pathway functions to remove protein aggregates/complexes, organelles, or viral particles, whereas the UPS system functions to degrade a single protein. Contrary to this general rule, GSNOR1 and Adi3 are selectively degraded through the autophagy pathway (Table 1) [92,94]. One logic question is whether this is economic for cells to do so energy-wise? It remains to be determined whether GSNOR1 and Adi3 can serve as cargo receptors to mediate the degradation of their interacting proteins and whether the degradation of a single protein via the selective autophagy pathway is a common phenomenon or just rare exceptions.

## 3. Selective Autophagy—A Battlefield between Plant–Pathogen Arms Race

The roles of selective autophagy in plant–pathogen interactions have been extensively reviewed recently [10,12,41,46,47]. Here, we just briefly review the arms race between plant and pathogens and focus our attention on the newest findings.

### 3.1. Selective Autophagy-Mediated Plant Immunity against Viruses

The degradation of viral proteins with key roles in viral virulence by selective autophagy plays critical anti-viral roles in plants. It has been shown that NBR1 binds viral proteins such as capsid protein P4 of Cauliflower mosaic virus (CaMV) and HCpro (Helper component proteinase) of TuMV, a potent viral suppressor of RNA silencing (VSR) or viral particles of CaMV and mediates their autophagic degradation, leading to the restriction of viral infection [55,95]. However, how NBR1 binds to P4 and HCpro is unknown. One possibility is that the HCpro is ubiquitinated before associating with NBR1 because the co-localization of NBR1 with HCpro is significantly reduced by the mutations in the UBA domain of NBR1 [55]. Tobacco calmodulin-like protein rgs-CaM may serve as a selective autophagy receptor for degradation of Cucumber mosaic virus (CMV) 2b, a potent VSR, to suppress host anti-viral defense [96]. Recently, Jiang et al. [97] identified a new cargo receptor *Nb*P3IP with a previously unknown function, which specifically interacts with the P3 protein (VSR) of Rice stripe virus (RSV) and *Nb*ATG8f. These interactions mediate the selective degradation of the P3 protein and limit RSV infection [97].

Beclin1/ATG6 in *Nicotiana benthamiana* selectively mediates the degradation of TuMV RNA-dependent RNA polymerase (NIb) in an ATG8a-dependent manner [98]. Beclin1/ATG6 interacts with NIb through the highly conserved GDD motif [98]. The loss of function of either Beclin1 or ATG8a enhances NIb accumulation and promotes viral infection. Conversely, the over-expression of either Beclin1 or ATG8a reduces NIb accumulation and inhibits viral infection [98]. This is the first report showing that an ATG protein (Beclin1/ATG6) functions as a selective cargo receptor in xenophagy.

The replication initiator protein C1 of a geminivirus, Tomato leaf curl Yuannan virus (TLCYnY) is localized in the nucleus. The interaction of C1 with *Nb*ATG8h leads to the translocation of the C1 protein from the nucleus to the cytoplasm and results in its degradation by selective autophagy in an AIM-dependent manner [99]. The nucleus-to-cytoplasm translocation of C1 is dependent on the exportin1 (XPO1)-mediated nuclear export pathway. However, the possibility of the newly synthesized C1 is degraded via the selective autophagy pathway before being targeted to the nucleus cannot be excluded, given that exportin1 is required for the nucleocytoplasmic transport of mRNA [100].

βC1 of Cotton leaf curl Multan virus (CLCuMuV)-associated Cotton leaf curl Multan betasatellite (CLCuMuB) is recruited to autophagosomes and subsequently degraded in vacuole through directly interacting with ATG8f and disruption of the βC1–ATG8f interaction resulted in an increased accumulation of viral DNA [101]. Cytosolic glyceraldehyde-3-phosphate dehydrogenases (GAPCs) negatively regulate autophagy and immunity through directly interacting with ATG3 [102]. Interestingly, βC1 directly interacts with GAPCs, and the interaction of βC1 with GAPCs disrupts the interaction of GAPCs with ATG3, leading to the de-repression of autophagy governed by GAPCs [103]. The point mutations within βC1 protein (βC13A) that impair GAPCs binding abolish the GAPCs–ATG3 interactions and fail to induce autophagy. As a result, the virus carrying mutant βC13A displayed increased symptoms and viral DNA accumulation associated with decreased autophagy in plants [103]. It seems that the host plant develops two different strategies to combat the viral infection through targeting the βC1 protein to the autophagy pathway. On one hand, βC1 is degraded directly by selective autophagy through interacting with ATG8f [101]. On the other hand, βC1 outcompetes ATG3 for GAPCs binding and thus release the autophagy-dependent immunity that is negatively regulated by GAPCs [103].

### 3.2. Pathogens Develop Various Strategies to Counteract the Host Defense Mediated by Selective Autophagy

Plant viruses develop various strategies to combat the host defense mediated by autophagy pathways. Viral proteins can interfere with or block the selective autophagy pathways either directly or indirectly. Viral proteins can also serve as cargo receptors to mediate the degradation of host proteins with anti-viral functions (Table 2).

TuMV VPg (Nuclear inclusion protein) and the small integral membrane protein 6K2 (6 kDa protein 2) antagonize host defense by blocking the NBR1-mediated autophagic degradation of HCpro [56]. In addition, TuMV VPg interacts with and mediates the degradation of REM1.2, a protein that negatively regulates the size exclusion limit (SEL) of plasmodesmata (PD), probably via both the 26S proteasome and autophagy pathways to facilitate the cell-to-cell movement of TuMV [72]. The S-acylation of *Nb*REM1.4/*Os*REM1.4 is required for their targeting to PD (104). The movement protein of RSV, NSsv4, interacts with the C-terminal domain of *Nb*REM1.4/*Os*REM1.4, and this interaction interferes with the S-acylation of *Nb*REM1/*Os*REM1.4 and results in the prevention of the PM targeting from ER. The non-acylated *Nb*REM1/*Os*REM1.4 sequestered at the ER is degraded through the autophagy pathway [104]. It is unclear whether NSsv4 serves as a receptor to mediate the specific degradation of *Nb*REM1/*Os*REM1.4.

Many viral proteins can function as cargo receptors to mediate the degradation of host proteins with anti-viral activities. TuMV VPg mediates the selective autophagic degradation of host SGS3 and RDR6, which are two key proteins in generating secondary siRNA and in the amplification of RNA silencing signals, through interacting with SGS3 [73]. Similarly, rgs-CaM induced by Tomato yellow leaf curl China virus (TYLCCNV) infection in *N. benthamiana* promotes TYLCCNV infection by interacting with SGS3 to mediate its autophagic degradation in *N. benthamiana* [105]. However, whether rgs-CaM functions as a cargo receptor is unclear. P0, a VSR from Turnip yellows virus (TuYV) triggers the degradation of Agronaute1 (AGO1), a key component of RNA-induced silencing complex (RISC), by the autophagy pathway [106]. Interestingly, Machaeli et al. [107] recently found ATI1/2 proteins present in the P0-induced ER structures. Since ATI1/2 interact with both ATG8 and AGO1 and P0 interacts with AGO1 [84,107,108,109], ATI1/2 likely serve as selective cargo receptors to mediate the autophagic degradation of the AGO1 together with P0 and ATI1/2, which is confirmed by the fact that P0-induced ER vesicles are targeted to the vacuole in an ATG5- and ATG7-deppendent manner [107]. As expected, ATI1/2 deficiency attenuates the P0-mediated decay of membrane-bound AGO1 and compromises post-transcriptional gene silencing [107].

It has been reported that Beclin1/ATG6 in *N. benthamiana* serves as a cargo receptor that selectively mediates the degradation of TuMV NIb protein (RDRP) and restricts TuMV replication [98]. Interestingly, the same group showed that NBR1 can serve as a selective receptor for TuMV NIb via interacting with ATG8f [110]. However, instead of being targeted to vacuoles for degradation, the NIb-NBR1-ATG8f-containing autophagosomes, to which the viral replication complexes (VRCs) are associated, are targeted to the tonoplast via an interaction between ATG8f and the tonoplast-intrinsic protein 1 (TIP1), leading to robust viral genome translation/replication and virion assembly in the tonoplast-associated VRCs [110]. It seems that the NBR1- and Beclin1-mediated selective autophagy of NIb antagonize with each other to promote and inhibit the TuMV infection, respectively [98,110].

The case studies for TuMV clearly show that the selective autophagy pathway is a battlefield for the armrace between host and viral pathogens. On one hand, the host concurrently targets multiple TuMV proteins (HCpro and NIb) by different receptors for autophagic degradation to ensure the suppression of viral infections with high efficiency. On the other hand, multiple TuMV proteins target different host factors with a defense role for autophagic degradation to facilitate its replication/movement or interfere with autophagic degradation of its own components or entire viral particles. In addition, the selective receptor/adaptor NBR1 plays both pro-viral and anti-viral roles depending on the different contexts and interacting with different ATG8 isoforms [98,110]. Furthermore, the NIb can be targeted by different selective receptors, resulting in a totally opposite effect on TuMV infectivity [98,110]. These results depict a vivid picture of an armrace between the host and virus, in which autophagy machinery is utilized for their own benefits (Table 2).

### 3.3. Pathogen Proteins Counteract Host Selective Autophagy-Mediated Defense through Competitive Binding with Selective Autophagy Components

In addition to the strategies mentioned in the last section, recent reports indicate that pathogens counteract host-selective autophagy-mediated defense through competing binding with selective autophagy components. γb, a VSR from Barley stripe mosaic virus (BSMV), directly competes with ATG8 for ATG7 binding, leading to the impaired association between ATG7 and ATG8 [111]. A single point mutation in γb abolishes its interaction with ATG7 in *N. benthamiana* as well as its ability to attenuate the anti-viral resistance conferred by the host autophagy. This study reveals that the BSMV γb protein subverts autophagy-mediated anti-viral defense by disrupting the ATG7–ATG8 interaction [111].

### 3.4. An Effector from Phytophthora Infestans, pexRD54, Serves as a Dual Selective Receptor to Suppress and Promote the Host Autophagy-Dependent Defense, Respectively

It has been shown that the RXLR (Arg-X-Leu-Arg)-type effector secreted from *Phytophthora infestans*, PexRD54, can suppress NBR1-mediated defense through out-competing NBR1 for ATG8CL binding [40]. PexRD54 contains an AIM and has a higher binding affinity for the ATG8CL than NBR1 and as a result, PexRD54 out-competes the NBR1 for ATG8CL binding and abrogates the NBR1-mediated degradation of defense-related cargoes [40] (Figure 2). Interestingly, during infection, the host–microbe interface is a hotspot for autophagosome biogenesis, and the autophagosomes are diverted toward the haustoria [112]. However, the reason behind this is not understood. Recently, it has been revealed that PexRD54 imitates starvation conditions and serves as a receptor for recruiting the small GTPase Rab8a with a role in basal resistance against *P. infestans*, and the lipid droplets (LDs) associated with Rab8a to form a distinct LDs-Rab8a-PexRD54-ATG8CL autophagosomes [113] (Figure 2). As a result, the Rab8a is trapped inside the autophagosomes and the Rab8a-mediated basal immunity is pacified [113] (Figure 2). Furthermore, instead of targeting to vacuole for degradation, the LDs-Rab8a-PexRD54-ATG8CL autophagosomes are diverted to the haustorial interface to supply the lipids required for the extrahaustorial membrane (EHM) and other nutrients engulfed in the autophagosome for the benefit of *P. infestans* [113] (Figure 2). These findings demonstrate that that the pathogen has evolved to create an effector to counteract the host defense via hijacking host autophagy machinery.

## 4. Post-Translational Modifications of Cargo Receptors Alter the ATG8s Accessibility or Binding Affinity

As autophagosome biogenesis and cargo recycling is energy-costing process, both cargo-to-SAR (selective autophagy receptor) and SAR-to-ATG8 interactions are tightly regulated by post-translational modifications [114]. Selective autophagy receptors are subjected to post-translational modifications adjacent to AIM for tightly interacting with ATG8. For example, human p62/SQSTM1 (Sequestosome 1), FUNDC1 (Fun14 domain containing 1), and Optineurin undergo phosphorylation that leads to a stronger interaction with ATG8s and facilitates cargo recruitment [115,116]. In the case of DSK2, the phosphorylation of DSK2 by BIN2 increases the interaction of BES1 with ATG8e, possibly through the exposure of its AIMs for ATG8 binding, and accelerates its autophagic degradation under drought conditions [67]. In response to SA or flg22 treatment, Mitogen-activated protein kinase 3 (MPK3) interacts with and phosphorylates Exo70B2, a subunit of the exocyst complex, which in turn results in the enhanced interaction of Exo70B2 with ATG8 [117]. The hypoxia-induced S-nitrosylation of GSNOR1 at Cys^10^ induces its conformational changes, enabling its normally buried AIM exposed and accessible for ATG8 binding [92]. For *At*TSPO–ATG8 interaction, it was speculated that heme binding to *At*TSPO may locally generate reactive oxygen species (ROS), leading to the oxidation of *At*TSPO. As a result, the oxidation of *At*TSPO induces a conformational change and makes its AIM-related motif accessible to ATG8 proteins. Alternatively, lipid peroxidation within the vicinity of *At*TSPO after the binding of heme may create a bilayer distortion, inducing *At*TSPO conformational changes and allowing the AIM motif to be exposed [69,70]. These results indicate that the post-translational modifications of cargo receptors are critical for the accessibility or binding affinity to ATG8s.

## 5. Identification of More ATG8-Interacting Proteins Using Newly Developed Software and Techniques (Proximity Tagging)

The receptors or adapters of selective autophagy are specified by the presence of short AIMs or a UIM that interacts with ATG8. Using more stringent criteria, Xie et al. [118] developed a bioinformatics approach, High-Fidelity AIM (hfAIM) (http://bioinformatics.psb.ugent.be/hfAIM/), to reliably identify AIMs in proteins. They demonstrate that the use of the hfAIM method allows for an in silico high-fidelity prediction of AIMs in AIM-containing proteins (ACPs) on a genome-wide scale in various organisms. By using hfAIM, they identify putative AIMs in the Arabidopsis proteome. They identified nine peroxisomal PEX proteins that contain hfAIM motifs and confirmed that *At*PEX6 and *At*PEX10 interact with ATG8 *in planta*. Mutations within or nearby the hfAIMs of these PEX genes resulted in defective growth and development in various organisms.

Y2H has been very successful in identifying ATG8-interacting proteins receptors/adapters or cargo proteins. Almost all the receptors/adapters or cargo proteins are identified using this approach [23,70,71,84,119]. However, using a BirA tag and proximity-dependent biotin identification (BioID) analysis [120], Macharia et al. [121] identified 67 proteins that interact with ATG8s from *N. benthamiana* plants infected by a fast replicating TMV strain. Sixteen of these proteins are known to interact with ATG8 or its orthologs in mammalian and yeast systems. The interacting proteins were categorized into four functional groups: immune system process, response to ROS, sulfur amino acid metabolism, and calcium signaling. Huntingtin-interacting protein K-like (HYPK) was validated as an ATG8-interacting protein using Y2H and BiFC (Bimolecular fluorescence complementation). With the development of more powerful tools such as TurboID-based proximity labeling [122], more ATG8-interacting proteins will be identified in the near future. The development of new bioinformatic tools and new techniques of studying protein–protein interactions will help to identify more biological processes that are regulated by selective autophagy.

## 6. Perspectives and Future Directions

Selective autophagy has been shown to play central roles in dealing with diverse stress conditions. Significant progress has been made in the past few years in understanding roles of selective autophagy in the clearance of cellular proteins or protein aggregates, damaged organelles, as well as invading pathogens. Using Y2H and Mass spectrometry (MS)-based proteomic approaches, many AIM- or UIM-containing proteins have been identified in plant genomes. We believe that the numbers will keep rising with the development of various advanced tools in the near future. The next challenges in the field will be the identification of the bona fide cargo receptors or adapters from these AIM- or UIM-containing proteins and the identification of the cargoes that interact with these newly identified selective receptors. High-throughput ATG8 isoform-specific interactome studies under different stress conditions and in different cell types/tissues will identify more such receptors. The identification of novel receptors and cargoes will greatly advance the understanding of the roles of selective autophagy and uncover the biological processes in which selective autophagy participates. Meanwhile, with more selective cargo receptors identified, more stringent or degenerative AIM or UIM domains will be uncovered, which in turn will help reliably and precisely predict the cargo receptors of selective autophagy. For the receptors that bear both AIM and UIM, it is possible that they selectively mediate the degradation of different cargoes under different conditions, in different cell types/tissues or at developmental stages. Future studies will certainly unveil these different possibilities.

The cargoes identified so far for the selective autophagy are almost exclusively recruited to the autophagosomes via interacting with receptors containing either an AIM or UIM in an ATG8-dependent manner. Identification of the ATG8-independent ways of cargo recruitment to the autophagosomes could be another challenge in the future. Manipulation of the components in the selective autophagy pathway will provide a promising means to improve crop adaptability to various stress conditions.

## Figures and Tables

**Figure 1 ijms-21-06321-f001:**
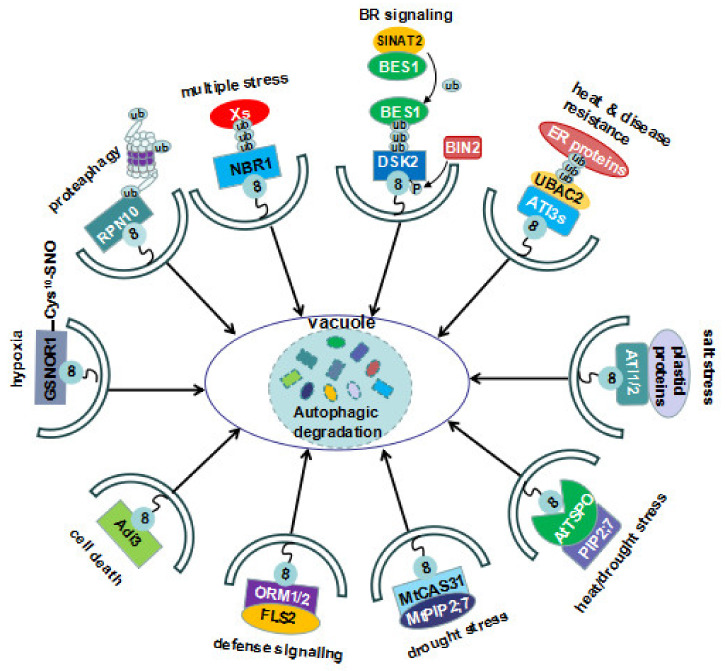
The cargoes and the receptors of selective autophagy identified in plants that are involved in various stress responses. Cys^10^–SNO represents the S-nitrosylation at Cys^10^ under hypoxia conditions. This modification induces the conformational changes of S-nitrosoglutathione reductase 1 (GSNOR1), enabling its ATG8-interacting motifs (AIM) to be exposed and accessible for ATG8 binding. Neighbor of BRCA1 (NBR1), Dominant suppressor of KAR 2 (DSK1), and Ubiquitin-associated domain2 (UBAC2) mediate the autophagic degradation of polyubiquitinated cargoes; three ubiquitin moieties represent polyubiquitination; Xs represents multiple polyubiquitinated proteins or protein aggregates that were induced under stress conditions; BRI1-EMS suppressor 1 (BES1) is firstly polyubiquitinated by Seven in absentia of Arabidopsis 2 (SINAT2), a Really interesting new gene (RING)-type E3 ligase, and the polyubiquitinated BES1 is recruited to the autophagosome by interacting with DSK2; DSK2 is phosphorylated at the sites flanking the two AIMs within its amino acid sequence. This modification results in the increased interaction between DSK2 and ATG8e.

**Figure 2 ijms-21-06321-f002:**
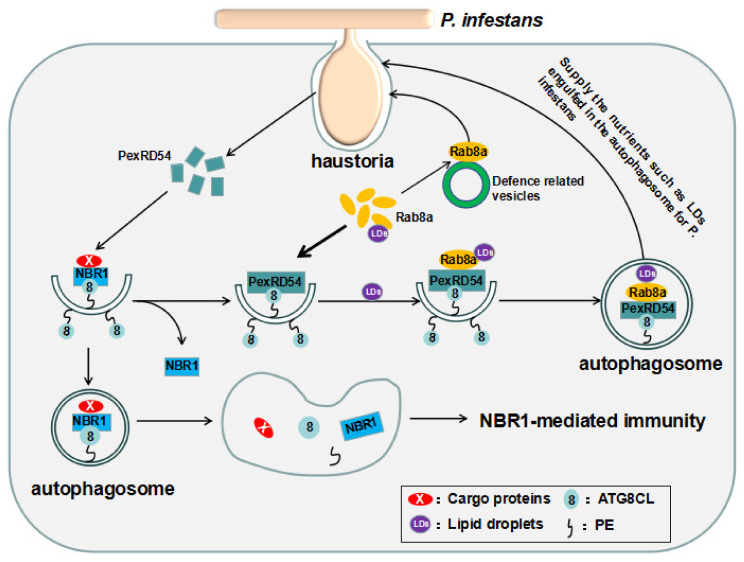
The effector secreted from *Phytophthora infestans*, PexRD54, functions as a receptor to counteract host defense in two different ways. RXLR (Arg-X-Leu-Arg)-type effector PexRD54 secreted from the haustotia of *P. infestans* has a higher affinity for binding ATG8CL than NBR1. NBR1 is out-competed and dispelled by PexRD54, and the NBR1-mediated immunity against *P. infestans* is abrogated. Meanwhile, PexRD54 mimics carbon starvation and induces the formation of the Rab8a/LDs-PexRD54-ATG8CL autophagosomes by directly and preferentially interacting with the inactive GDP-bound form of Rab8a, a host vesicle transport regulator, and ATG8CL, respectively, and recruiting Rab8a-associated lipid droplets (LDs). As a result, the basal immunity mediated by Rab8a against *P. infestans is* pacified by trapping the Rab8a in the autophagosomes. Instead of targeting to vacuoles for autophagic degradation, the Rab8a/LDs-PexRD54-ATG8CL autophagosomes are diverted to haustoria, and the cargoes engulfed in the autophagosomes, such as LDs, could be a source of the lipids for the extrahaustorial membrane (EHM) of the haustoria. PexRD54 not only interferes with the NBR1- and Rab8a-mediated resistance but also supplies lipids or other materials for the parasite.

**Table 1 ijms-21-06321-t001:** The receptors and cargoes of selective autophagy and their functions in plants.

Receptors/Adapters	Cargos	ATG8 Isoforms	Functions	References
*At*RPN10	Proteasome	*At*ATG8e (UIM)	Proteaphagy	[21]
*At*ATI3s *At*UBAC2	Ubiquitinated ER proteins	*At*ATG8a and 8f	Heat tolerance and disease resistance	[23]
*At*NBR1	Ubiquitinated proteins	*At*ATG8s	Clearing misfolded proteins, protein aggregates and pathogens proteins, or particles induced under stress conditions	[25,55,95]
*At*DSK2	*At*BES1	*At*ATG8e	BR signaling and stress tolerance	[66]
*At*TSPO	*At*PIP2;7	*At*ATG8?	Drought tolerance	[69,70]
*Mt*CAS31	*Mt*PIP2;7	*Mt*ATG8a	Drought tolerance	[71]
*At*ORM1/2	*At*FLS2	*At*ATG8a, 8d, 8e, 8i	Negative regulate *At*FLS2-mediated defense	[81]
*At*ATI1/2	Plastid proteins	*At*ATG8f *At*ATG8h	Salt stress tolerance; Chlorophagy and plastid proteins degradation	[85,86]
*At*GSNOR1	*At*GSNOR1	*At*ATG8	Hypoxia responses	[92]
*Sl*Adi3	*Sl*Adi3	*Sl*ATG8h	Cell death and disease resistance	[94]

**Abbreviations: Plant species:** At, *Arabidopsis thaliana*; Nb, *Nicotiana benthamiana*; Mt, *Medicago truncatula*; Sl, *Solanum lycopersicum*. **Proteins:** Adi3, AvrPto-dependent Pto-interacting protein 3; ATI, ATG8-interacting protein; BES1, BRI-EMS suppressor 1; BR, Brassinosteroid; CAS31, Cold-acclimation-specific 31; DSK2, Dominant suppressor KAR 2; FLS2, Flagellin-sensing 2; GAPC, Glyceraldehyde-3-phosphate-dehydrogenase; GSNOR1, S-nitrosoglutathione reductase; HCpro, Helper component proteinase; NBR1, Neighbor of BRCA 1; ORM1, Orosomucoid 1; PIP2;7, Plasma membrane intrinsic protein 2;7; RPN10, 26S proteasome regulatory particle; TSPO, Tryptophan-rich sensory protein/translocator; UBAC2; Ubiquitin-associated domain 2; UIM, Ubiquitin-interacting motif; Question mark indicates “unknown” ATG8 isoform.

**Table 2 ijms-21-06321-t002:** Selective autophagy involved in plant–pathogen interactions.

Pathogens	Pathogen Proteins	Host Proteins	Functions	References
**Anti-viral functions**				
Caulimovirus CaMV	P4	*At*NBR1; *At*ATG8a	Selective degradation of P4	[55]
Potyvirus TuMV	HCpro	*At*NBR1; *At*ATG8a	Selective degradation of HCpro	[95]
CucumovirusCMV	2b	rgs-CaM; ATG8	Selective degradation of 2b and rgsCaM	[96]
Potyvirus RSV	p3	*Nb*P3IP; *Nb*ATG8f	Selective degradation of P3	[97]
Potyvirus TuMV	NIb	*Nb*Beclin1 *Nb*ATG8a	Selective degradation of NIb	[98]
Geminivirus TLCYnY	C1	*Nb*ATG8h	Selective degradation of C1	[99]
Geminivirus CLCuMuB	βC1	*Nb*ATG8f	Selective degradation of βC1	[101]
**Pro-pathogen functions**				
*Phytophthora infestans*	PexRD54	*St*ATG8CL	PexRD54 outcompetes the NBR1 for ATG8CL binding and counteracts NBR1-mediated host defense.	[40]
Potyvirus TuMV	Vpg	*At*REM1.2, *At*SGS3, *At*RDR6	Antagonize *At*REM1 function and promote the cell-to-cell movement of TuMV. Suppress host anti-viral RNA silencing pathway.	[72,73]
Geminivirus CLCuMuB	βC1	*Nb*GAPCs	βC1 out-competes ATG3 for GAPCs binding and disrupts GAPCs-mediated immunity.	[103]
Potyvirus RSV	NSsv4	*Nb*REM1/*Os*REM1	Inhibit plasmodesmata targeting of REM1, trigger the degradation of the non-acylated REM1 and promote viral cell-to-cell movement.	[104]
Polerovirus TuYV	P0	*At*AGO1, *At*ATI1/2	Suppress host anti-viral RNA silencing pathway.	[106,107]
Potyvirus TuMV	NIb	*Nb*NBR1/*At*NBR1 *Nb*ATG8f3/*At*ATG8f *Nb*TIP1	Targeting TuMV VRCs to tonoplast and promote viral replication and assembly.	[110]
Hordeivirus BSMV	γb	*Nb*ATG7	γb out-competes the ATG8 for ATG7 binding and compromises the autophagy-mediated defense.	[111]
*Phytophthora infestans*	PexRD54	*St*Rab8a	Pacify *St*Rab8a-mediated defense and/or supply nutrients (lipid droplets) for haustoria of *P. infestans*.	[112,113]

**Abbreviations: Plant species:***At*, *Arabidopsis thaliana*; *Nb*, *Nicotiana benthamiana*; *Os*, *Oryza sativa*; *St*, *Solanum tuberosum*. **Viruses:** BSMV, barley stripe mosaic virus; CaMV, cauliflower mosaic virus; CLCuMuV, cotton leaf curl Multan virus; CMV, cucumber mosaic virus; RSV, rice stripe virus; TLCYnV, tomato leaf curl Yunnan virus; TuMV, turnip mosaic virus; TuYV, Turnip yellows virus. **Proteins:** GAPCs, cytosolic glyceraldehyde-3-phosphate dehydrogenases; HCPro, helper-component proteinase; NBR1, Neighbor of BRCA1; Rab8a, P3IP, P3-interacting protein; Ras-related protein 8a; REM, remorin; rgs-CAM, calmodulin-related protein; VPg, Viral protein genome-linked; VSR, viral suppressor of RNA silencing.

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
