# Peer review of "Emerging Roles of the Selective Autophagy in Plant Immunity and Stress Tolerance"

_ijms, 2020, doi:10.3390/ijms21176321_

Round 1

Reviewer 1 Report

The manuscript requires some minor modifications as follows.

1. Lines 41-42; "The AIM forms hydrophobic bonds with two conserved hydrophobic pockets of the AIM-docking sites (ADS) on ATGs." I am not sure about this since the paper authors referred (Stephani & Dagdas, J. Mol. Biol. 2020 432; 63-79) does not include the statement. Please modify or refer to the correct paper.

2. Lines 50-52; "It has been demonstrated that the selective autophagy plays central roles in removing aggregated proteins and protein aggregates." There are two similar words at the end of the sentence.

3. Line 53; "peroxiphagy" and Line 59 "perophagy" should be better with "pexophagy". Line 59; "clorophagy" should be "chlorophagy".

4. Lines 122; "Arabidopsis growth and development and trades for enhanced stress responses." Meaning of the sentences unclear.

5. Line 343; "ER bodies" should be "ER structures".

6. Line 381; "P. infestans" should be better with "Phytophthora infestans" because this is the first appearance in the text.

7. Table 1 and Table 2; It is better to add plant species in the column.

Author Response

Responses to Reviewer 1:

The manuscript requires some minor modifications as follows.

  1. Lines 41-42; "The AIM forms hydrophobic bonds with two conserved hydrophobic pockets of the AIM-docking sites (ADS) on ATGs." I am not sure about this since the paper authors referred (Stephani & Dagdas, J. Mol. Biol. 2020 432; 63-79) does not include the statement. Please modify or refer to the correct paper.

Thank reviewer 1 pointed this out! We have double checked the original sources and cited the correct references. Accordingly, we have made changes to the reference list as well as the citations in Table 1 and Table 2.

  1. Lines 50-52; "It has been demonstrated that the selective autophagy plays central roles in removing aggregated proteins and protein aggregates." There are two similar words at the end of the sentence.

We have removed “aggregated proteins and”.

  1. Line 53; "peroxiphagy" and Line 59 "perophagy" should be better with "pexophagy". Line 59; "clorophagy" should be "chlorophagy".

Thank reviewer 1 for catching the errors! We have made the changes.

  1. Lines 122; "Arabidopsis growth and development and trades for enhanced stress responses." Meaning of the sentences unclear.

We have changed the sentence "inhibits Arabidopsis growth and development and trades for enhanced stress responses." to “inhibits Arabidopsis growth/development and activates stress responses.”

  1. Line 343; "ER bodies" should be "ER structures".

Thank the reviewer 1 for the correct term! We have made the changes (line 347).

  1. Line 381; "P. infestans" should be better with "Phytophthora infestans" because this is the first appearance in the text.

We have made the changes as the reviewer 1 suggested (line 347).

  1. Table 1 and Table 2; It is better to add plant species in the column.

This is really a good suggestion. Due to the space limitation, it is difficult to add a column without impacting the overall layout of the tables. As an alternative, we have added footnotes for Table 1 and Table 2 with plant species information included.

Reviewer 2 Report

Overall, I believe this is a good review regarding cross-talk between the UPS and autophagy pathways, with an interesting emphasis on selective-autophagy cargo receptors, notably in the context of plant-pathogen interactions.

I believe that the review is overall concise enough and well segmented to enable the reader to find relevant information.

Nevertheless, some minor corrections should be made :

Line 16: Delete “the” before "selective autophagy"

Lines 81-82: “insoluble detergent-resistant” is redundant, rephrasing should be performed

Lines 120-122: rephrase “which inhibits Arabidopsis growth and development and trades for enhanced stress responses.”

Line 159: should be: the “26S proteasome”

Line 166: It should not be “Instead” but “In addition”.

Lines 182-183: “were found to be interacted with” is not a correct phrase

Line 325: should be “SEL” for size-exclusion limit and not “ESL”; also to be changed in the abbreviation list

Lines 414-416: the acronym “SAR” has not been introduced

These suggested corrections are not exhaustive, so I would recommend that the manuscript be corrected by a native English speaker.

Author Response

Responses to Reviewer 2:

Overall, I believe this is a good review regarding cross-talk between the UPS and autophagy pathways, with an interesting emphasis on selective-autophagy cargo receptors, notably in the context of plant-pathogen interactions.

Thank reviewer 2 for the positive comments!

I believe that the review is overall concise enough and well segmented to enable the reader to find relevant information.

Thank reviewer 2 for the positive comments!

Nevertheless, some minor corrections should be made :

Line 16: Delete “the” before "selective autophagy"

Thank reviewer 2 for catching the error. We have deleted “the”.

Lines 81-82: “insoluble detergent-resistant” is redundant, rephrasing should be performed

We have deleted detergent-resistant” as reviewer 2 suggested.

Lines 120-122: rephrase “which inhibits Arabidopsis growth and development and trades for enhanced stress responses.”

We have changed the sentence "inhibits Arabidopsis growth and development and trades for enhanced stress responses." to “inhibits Arabidopsis growth/development and activates stress responses.”

Line 159: should be: the “26S proteasome”

Thank reviewer 2 for catching the error. We have made the correction.

Line 166: It should not be “Instead” but “In addition”.

We have made the change as reviewer 2 suggested.

Lines 182-183: “were found to be interacted with” is not a correct phrase

We have changed the sentence to“were found interacting with”.

Line 325: should be “SEL” for size-exclusion limit and not “ESL”; also to be changed in the abbreviation list

Thank reviewer 1 for catching the error. We have made change both in text and abbreviation list.

Lines 414-416: the acronym “SAR” has not been introduced

Thank reviewer 2 for pointing this out! We have spelled out the SAR as “selective autophagy receptor” in the text.

These suggested corrections are not exhaustive, so I would recommend that the manuscript be corrected by a native English speaker.

This manuscript has been thoroughly edited by a native English speaker.